# Alcohol, Cannabis and Crossfading: Concerns for COVID-19 Disease Severity

**DOI:** 10.3390/biology10080779

**Published:** 2021-08-16

**Authors:** Vijay Sivaraman, Morgan M. Richey, ABM Nasir

**Affiliations:** 1Department of Biological and Biomedical Sciences, North Carolina Central University, Durham, NC 27707, USA; 2Julius Chambers Biomedical and Biotechnological Research Institute, North Carolina Central University, Durham, NC 27707, USA; 3Department of Epidemiology, Gillings School of Global Public Health, UNC Chapel Hill, Chapel Hill, NC 27514, USA; mrichey@email.unc.edu; 4Business School, North Carolina Central University, Durham, NC 27707, USA; anasir@nccu.edu

**Keywords:** alcohol, cannabis, crossfading, pneumonia, COVID-19

## Abstract

**Simple Summary:**

Heavy alcohol and cannabinoid consumption prior to pneumonia infection resulted in an increased severity of disease in mice. Analysis of hospital discharge data showed an increased length of stay among alcohol-consuming pneumonia patients. Increased alcohol and cannabis sales during the early months of 2020 suggest that consumption of both substances has increased significantly during the pandemic, potentially representing an increased risk of severe COVID-19 mediated by lung inflammation.

**Abstract:**

Risk factors for severe COVID-19 pathology are currently being investigated worldwide. The emergence of this highly infectious respiratory disease has plagued the world, with varying severity across populations of different age, race, and socio-economic level. These data suggest that other environmental or social factors may contribute to this disease’s severity. Using a mouse model, we identify heavy alcohol and cannabinoid consumption as risk factors for increased pulmonary pathology in the setting of exposure to a microbial pulmonary pathogen (*K. pneumoniae*). We present observational evidence that pneumonia patients admitted to North Carolina hospitals have longer lengths of stay when they endorse alcohol use or have conditions considered alcohol attributable. We are concerned that the observed increase in alcohol and legal cannabinoid sales during lockdown and quarantine may contribute to increased pulmonary pathology among patients who become infected with COVID-19.

## 1. Introduction

It is well accepted in the literature that heavy alcohol usage has been associated with multiple pathologies, ranging from gastrointestinal and renal toxicity to neurological dysfunction [1,2,3]. The evidence supporting a relationship between alcohol abuse and more severe pulmonary disease is also documented in the medical literature [4]. Susceptibility to pulmonary infection has been associated with alcohol use, as incidence of both community-acquired and hospital-acquired pneumonias (CAP/HAP) have been observed to be more common among those who abuse alcohol. Alcohol abuse is also associated with pneumonia outcomes, including longer recovery times, more severe bacteremia, and increased mortality [5]. Further evidence suggests that the severity of ARDS and pneumonia is greatly increased in those who have a history of alcohol and polysubstance abuse [4,6].

Cannabis use has also been associated with pulmonary dysfunction, such as wheezing, shortness of breath, and cough [7,8,9]. However, restrictions on the study of controlled substances and challenges in study design have resulted in conflicting evidence of a causative relationship between cannabis use and frank pulmonary disease [10,11,12,13]. Similar to pathologies associated with alcohol abuse, cannabis abuse is associated with various pulmonary pathologies such as bronchial damage, basal cell hyperplasia, and increased frequency and severity of asthma exacerbations [12,14]. The practice of consuming alcohol and cannabis simultaneously, colloquially known as “crossfading”, is known to be common among cannabis users, and has only recently been a target of investigation [15]. Recent study into the effect of simultaneous cannabis and alcohol consumption suggests that cannabis modulates the pulmonary inflammatory effects of alcohol via toll-like receptors (TLRs) that sense pathogens and initiate inflammatory responses [16]. Despite these published findings, many states have legalized medicinal and recreational cannabis usage, highlighting the need to understand the effects of simultaneous alcohol and cannabis usage [17,18,19].

The novel coronavirus SARS-CoV-2 has presented as a highly communicable respiratory virus and the causative agent for the lethal pulmonary disease known as COVID-19. SARS-CoV-2 can easily be transmitted between people in close proximity through respiratory droplets or microscopic aerosolized particles. As of 1 March 2021, over 2.5 million fatalities associated with COVID-19 have occurred throughout the world, with the majority occurring in the United States, India, Brazil, Russia, and Western Europe (United Kingdom, Italy, Spain, and France) (https://coronavirus.jhu.edu/map.html (accessed on 1 March 2021). While severe pathology and disease initially was thought to primarily affect older adults with various pre-existing health comorbidities, otherwise healthy individuals have also died from the disease, and the determinants of susceptibility are still being investigated. These data suggest that environmental and social factors such as substance use may contribute to the susceptibility to severe disease among those infected with SARS-CoV2. It is concerning that as the SARS CoV-2 pandemic continues to spread, excessive alcohol and cannabis consumption, possibly associated with the stress of isolation, quarantine and social distancing, may increase the risk of severe COVID-19 disease. The aim of this study is to evaluate risk factors associated with alcohol and cannabis consumption that may contribute to worsened COVID-19 disease prognosis and hospitalization.

## 2. Results

Our results suggest evidence that excessive alcohol consumption exacerbates pulmonary inflammation in a mouse model, and is associated with increased incidence of pneumonia, disease severity, and mortality due to a microbial infection [20]. To model this disease in vivo, we exposed 10 adolescent mice (C57-Bl6) to binge-level quantities of ethanol (5 mg/kg of body weight, day on/day off over 10 days), allowed a two-week washout period, and then challenged them with a sub-lethal intranasal dose of *K. pneumoniae* [20] (a common nosocomial pulmonary infectious microbe). Our results demonstrate more severe pathology (increased neutrophilic influx within alveolar space demonstrated by H&E staining, Figure 1) in the lungs of alcohol-exposed animals, compared to control mice exposed only to *K. pneumoniae*.

Experiments were performed to evaluate the mechanism behind alcohol-mediated pulmonary exacerbation, and these data have suggested a role for cannabinoid receptor-dependent signaling cascades leading to this response (in preparation). As a result, adolescent animals were exposed to either alcohol, an endogenous cannabinoid (WIN 55,212), or both, and then inoculated with *K. pneumoniae* intranasally (as previously described). These experiments yielded similar results to alcohol exposure (Figure 1). Cannabinoid-exposed mice demonstrated more severe disease compared to mock infected animals, as evidenced by increased inflammatory cytokines within pulmonary tissue, clinical signs of disease (increased respiration rate, hunching, and ruffling) and increased inflammatory pathology (neutrophilic influx, vascular edema). Interestingly, the mice that received both cannabinoids and alcohol had the most extensive neutrophilic influx, suggesting amplified effects by the two substances. The levels of inflammatory cytokine proteins IL-6, TNF-alpha and Mip1-alpha (common measures of inflammation) were assessed in each group via bronchioalveolar lavage fluid of euthanized mice. Both singly exposed groups of mice (alcohol, cannabinoids) displayed significantly higher levels of these cytokines compared to control mice, while the doubly exposed group (alcohol and cannabinoids) displayed the highest levels of the inflammatory proteins. These results are consistent (while not conclusive) with our hypothesis that cannabinoid signaling cascades contribute to severe pulmonary disease. While elevation of IL-6 and other inflammatory biomarkers may operate through CB1-independent pathways, these results suggest a role both for alcohol alone and the combination of alcohol and cannabis in increased morbidity and mortality in the setting of infection by a pulmonary pathogen. While these preliminary results are currently applicable only to mice, the possibility that similar mechanisms are present in humans should be investigated.

Confirmation of these results in subsequent studies could potentially prompt investigation of these pathways in humans. To evaluate the relationship between alcohol exposure and severe disease in humans, a retrospective cohort study of inpatient hospitalization in North Carolina was performed. A large North Carolina hospital discharge database, which contains records for all non-federal hospital discharges in the state, was leveraged for this study. During 2016–2017, 14,685 patients aged 18–75 were admitted with a primary diagnosis of pneumonia to North Carolina hospitals. These patients were stratified into two groups: one which endorsed alcohol use or had a history of disease considered 100% attributable to alcohol (N = 738), and a second group with no history of alcohol-related disease or endorsement of alcohol use (N = 14,127). Multivariable linear and Poisson regression models were employed to estimate the differences in length of stay between the two groups, and adjusted for relevant confounders (age, sex, race). Patients in the alcohol group stayed a model-estimated 1.1 days longer (95% confidence interval: 1.0, 1.2 days *p* < 0.001) in the hospital compared to the non-alcohol group (Figure 2). This small but significant increase in length of stay is similar in magnitude with other studies of alcohol as a risk factor for contracting pneumonia requiring hospitalization [21,22], inpatient length of stay [5,23,24,25], the necessity of mechanical ventilation, and incidence of acute respiratory distress syndrome (ARDS) [26]. While our results are consistent with other studies in the literature, cautious interpretation of the validity of our estimate is warranted due to limitations in controlling confounding in hospital claims data, and the possibility of inconsistent or deliberate non-disclosure of alcohol use by hospital patients.

By March and April 2020, much of the US enacted “Shelter-at-Home” policies and encouraged social distancing to curtail the spread of the SARS-CoV2 virus. In many states, businesses such as ABC stores and cannabis dispensaries, (in 9 of 11 states where legal) were deemed “necessary” and remained open for business. US alcohol sales abruptly increased by 21.8% in March 2020 compared to the same month in 2019 (Figure 3A). As the panic subsided, alcohol sales dropped slightly by 13.2% in April, followed by another increase by 22.9% in May, and remained significantly higher through November of 2020. These results were robust to controlling for time trend, and while some consumers may have been stockpiling alcohol, we hypothesize that increased sales during this time also correlated with increased consumption.

Figure 3B illustrates the dramatic increase in retail sales of marijuana during the SARS-CoV2 outbreak in four U.S. states during March 2020. According to the report by Marijuana Business Daily, the sales of recreational marijuana in California increased by 159% compared to the sales reported on the same day in 2019, while sales in Washington and Colorado increased by 100 and 46%, respectively. The SARS-CoV2 outbreak seemed to contribute to the spike in cannabis sales as average year-over-year sales on Mondays between 6 January and 9 March were up by 71, 10, and 15%, respectively, in California, Colorado, and Washington (Figure 3B).

As observed from self-reported data across the U.S., the cannabis usage rates during the COVID-19 outbreak increased in 2020. Usage rate is defined as the frequency of daily, weekly, and monthly uses (Figure 3B). A survey was conducted by Brightfield Group in March 2020, finding that the heaviest cannabis users reported intentions to increase their usage during the coronavirus outbreak, and that the proportion of respondents who used cannabis multiple times per day increased by 32%. These data do not consider the likely dramatic increase in sales of both illicit and legal cannabis.

## 3. Discussion

Our data suggest that alcohol and cannabis use may contribute to synergistic priming of more severe inflammation within the lung, in the setting of a pulmonary infection, and further investigation in humans may be warranted. Our findings using in vivo mouse models, the clinical literature, and population-based data all support the hypothesis that consumption of alcohol is associated with contracting pneumonia and experiencing protracted illness. The observed increase in alcohol and cannabis sales during the pandemic may constitute a risk factor for contracting SARS-CoV2 infection, or severe disease. Though the increased purchase and consumption of substances were likely coping strategies against the isolation and stress associated with the pandemic, the unappreciated effects upon pulmonary health are a real risk. It is of great concern that as our population attempted to minimize their likelihood of contracting COVID-19 by isolating and distancing, they may have been consuming compounds (and still consuming) that may increase their vulnerability to severe disease. Investigating the mechanisms involved in both alcohol and cannabis-mediated inflammatory priming within the lung is important to better understand the pulmonary immune system and may reveal pathways susceptible to pharmacological intervention. In conclusion, although the experimental data are emerging, the increase in alcohol and cannabis sales during the time of pandemic should focus the attention of health and scientific professionals to this potential problem, for both health of both direct and indirect victims of COVID-19.

## Figures and Tables

**Figure 1 biology-10-00779-f001:**
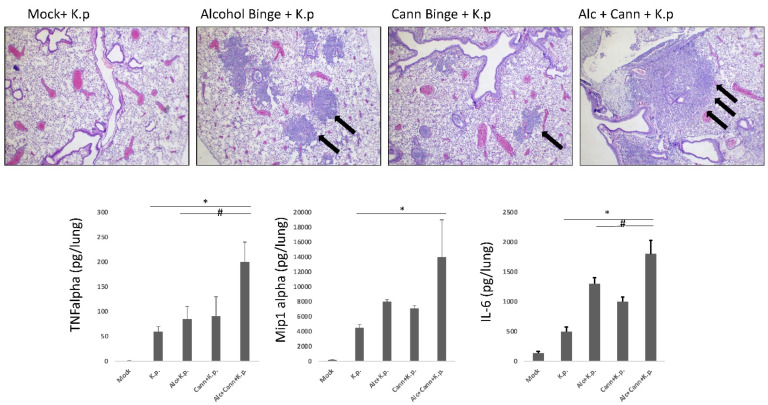
Mouse lung histopathology and inflammatory protein secretion. Increased neutrophilic influx and damage is present in lungs of mice exposed to substances prior to pneumonia challenge. Formalin fixed mouse lung sections are stained with H&E and magnified by 40×. Cellular influx and consolidation are demonstrated by purple staining, and inflammation is indicated with arrows. Inflammatory cytokine secretion measured from bronchioalveolar lavage fluid, using IL-6, TNF alpha and Mip1-alpha ELISAs (BD Bioscience). # indicates significant increase over single substance + K.p., * indicates significance over K.p. alone. K.p: *Klebsiella pneumoniae*, cann: cannabinoids, alc: alcohol.

**Figure 2 biology-10-00779-f002:**
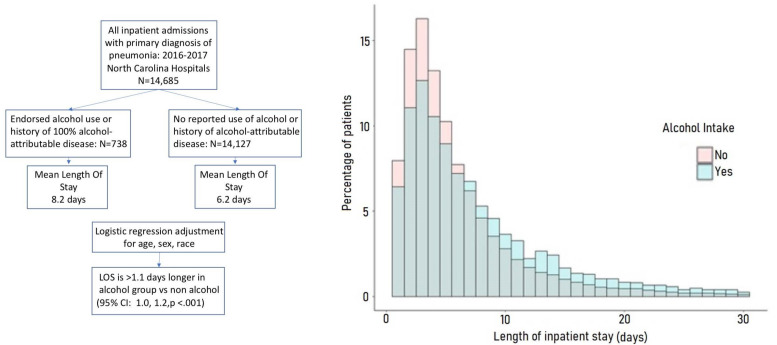
NC Hospital discharge data, 2016–2017. Patients in NC admitted for pneumonia with prior exposure to alcohol had extended stays, compared to those who reported no prior exposure. Multivariable linear and Poisson regression models adjusting for age, sex, and race were employed to estimate length of stay differences between the two groups.

**Figure 3 biology-10-00779-f003:**
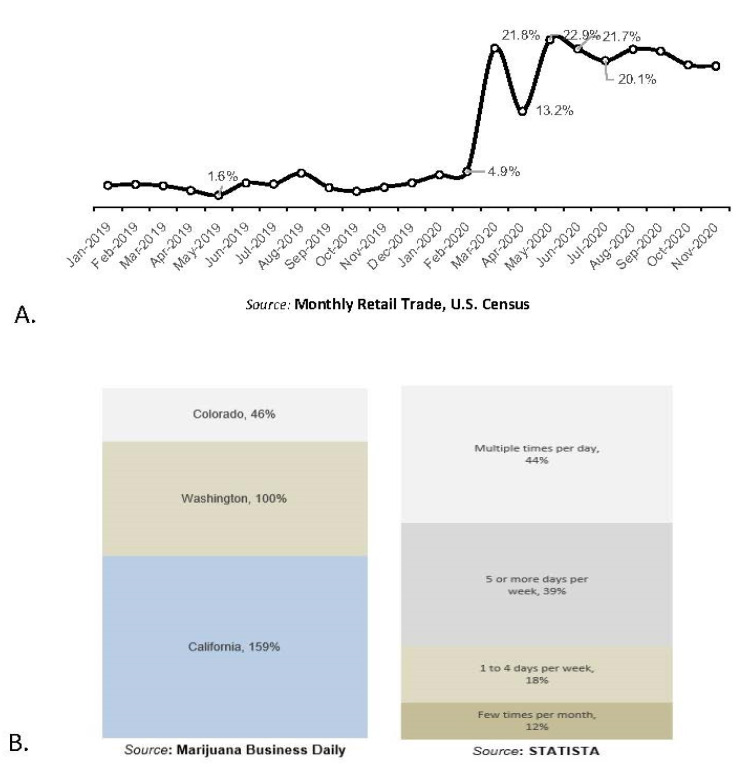
(**A**): The Monthly Retail Trade statistics data. Data are published by the U.S. Census (https://www.census.gov/retail/index.html#mrts (accessed on 1 March 2021)) and analyzed for the period 1 January 2019–1 November 2020. The U.S. Census reports the data as the ‘Advanced Monthly Sales for Retail Trade’, released monthly and seasonally adjusted. Panic-buying demonstrated by “Sales growth = [0.004]^(**) + [0.17]^* dummy; adj_(R^2) = 0.95”. Significance levels are reported as alpha 1, 5, and 10%. At 1% level of significance, confidence level is 99%. At 5% significance, confidence level is 95%. (**B**): Data obtained for Colorado, Washington, and California. See the link: (https://mjbizdaily.com/recreational-cannabis-retailers-sales-surge-during-coronavirus-pandemic/ (accessed on 1 March 2021). Data obtained from STATISA.com, based on analysis from Brightfield Group. See the link: (https://www.statista.com/statistics/1120186/increased-cannabis-use-during-coronavirus-outbreak-by-current-usage-us/ (accessed on 1 March 2021).

## Data Availability

NC Hospital discharge data are available with permission from the Cecil G. Sheps Center for Health Services Research.

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
