# Peer review of "Alcohol, Cannabis and Crossfading: Concerns for COVID-19 Disease Severity"

_biology, 2021, doi:10.3390/biology10080779_

Round 1

Reviewer 1 Report

The manuscript entitled "Alcohol, Cannabis and Crossfading: Concerns for COVID-19 2 Disease Severity". Title, abstract and overall rationale of work to some extent is good. However, there are still some minor concerns, which needs to be addressed and needs minor revision.

1) I would suggest the authors to enhance your theoretical discussion and arrives your debate or argument.

2) Author why only measured IL-6 cytokine and what about the other important cytokines that are play significant role in pulmonary pathology and COVID-19 disease. I think, author also need to demonstrate the role of IFN-g, TNF-a, IL-2, IL-10 theses cytokines.

3) Author must be elaborated the conclusion section and also write about the future prospective of this work.

4) Figure 1A and 1D must be increased resolution at least 300 dpi.

5) A flowchart should be added to this article to show the clear methodology.

Author Response

Dear Reviewer,  we appreciate your reading and interest in our work. Our responses to your comments are as follows:

1) We respect the reviewers concern to elaborate on our theoretical discussion, and have added text in response.

2) We agree that more inflammatory protein data would be useful, so have included cytokines TNF-alpha and Mip1-alpha data to the figure.

3) We have added text to discuss conclusions and future directions.

4) We have improved the resolution of figures to address this comment.

5) We have included a flow chart to explain how Figure 2 was generated from patient discharge data

Reviewer 2 Report

Even if the work has undisputed points of innovation, it must be completely rewritten because in my opinion it does not have the minimum acceptability requirements for the journal.

Author Response

We greatly appreciate the reviewer’s approval of our innovation.

We were surprised that they found our work to not meet the standards of this journal. As a perspective piece written during the emergence of the pandemic, we have combined the research interests of three diverse scientists to culminate with this short statement of concern regarding societal habits and downstream repercussions during the time of COVID-19.  We respectfully disagree with this reviewer’s view.